# Analysis of Indirect Costs of Absence Associated with Mental Disorders on the Basis of Social Security Data (2012–2023)

**DOI:** 10.3390/healthcare12171784

**Published:** 2024-09-06

**Authors:** Paweł Juraszek, Karolina Sobczyk, Karolina Krupa-Kotara, Mateusz Grajek

**Affiliations:** 1Department of Public Health, Faculty of Public Health, Medical University of Silesia in Katowice, 40-007 Katowice, Poland; pawel.s.juraszek@gmail.com; 2Department of Health Economics and Management, Faculty of Public Health, Medical University of Silesia in Katowice, 40-007 Katowice, Poland; ksobczyk@sum.edu.pl; 3Department of Epidemiology, Faculty of Public Health, Medical University of Silesia in Katowice, 40-007 Katowice, Poland; kkrupa@sum.edu.pl

**Keywords:** mental health, indirect costs, absenteeism, social insurance institution, disability pensions

## Abstract

Background: Mental and behavioral disorders significantly impair psychophysical functioning, leading to challenges in daily activities. The increasing recognition of the importance of mental health in global development goals has resulted in its inclusion in the United Nations’ Sustainable Development Goals. The burden of mental disorders has grown worldwide due to demographic changes, with substantial economic and social impacts. Objective: This study aimed to examine the indirect costs of mental disorders in Poland by analyzing the expenditures by the Social Insurance Institution (ZUS) on work incapacity benefits and disability pensions from 2012 to 2023. The goal was to identify trends, dependencies, and the economic impact of policy changes. Material and Methods: Data were collected from ZUS reports on annual expenditures for work incapacity benefits and disability pensions. Advanced statistical methods, including linear regression and Pearson correlation, were employed to analyze trends and relationships. Student’s *t*-tests assessed the statistical significance of the observed trends. Results: The expenditures on benefits for work incapacity due to mental disorders increased significantly over the past decade, particularly from 2020 to 2023, partly due to the COVID-19 pandemic. Short-term absenteeism due to mental disorders accounted for 10.5% of the total sick leave days. A strong positive correlation was found between years and benefit expenditures. Conversely, the disability pension expenditures exhibited a downward trend, potentially reflecting improvements in public health or policy changes. Conclusions: The significant increase in expenditures on benefits related to mental disorders may reflect greater awareness, improved diagnostics, and the impact of the COVID-19 pandemic. In contrast, the decline in disability pension expenditures could suggest improved mental health or effective policy measures. However, it is important to emphasize that the presented data are not the only factor influencing this situation. Multiple variables, including societal, economic, and healthcare system changes, contribute to these trends. Therefore, further research is necessary to fully understand the underlying causes and to guide effective policy development. Regular monitoring and continued investment in mental health are essential to managing indirect costs such as absenteeism and presenteeism efficiently.

## 1. Introduction

Mental and behavioral disorders belong to a group of health problems that lead to a reduction in the body’s psychophysical performance, making it difficult or impossible for individuals to function normally. In recent years, there has been increasing recognition of the invaluable role that mental health plays in achieving global development goals, which has translated into the inclusion of this area of health in the Sustainable Development Goals (SDGs) [1]. This United Nations (UN) global strategy includes 17 integrated Sustainable Development Goals, and it is recognized that action taken in the area of one goal will have an impact on outcomes in the areas of the other stated goals. The inclusion of mental health in the strategy recognizes its relevance, along with other important areas of human life, and that it is essential to ensure that, according to the document’s central tenet, all people enjoy peace and prosperity by 2030 [2].

The burden of mental and behavioral disorders is growing worldwide. Largely due to demographic changes, there has been an increase of several percent in the prevalence of disease entities in this disease group over the past decade. About 20 percent of children and adolescents worldwide have mental health problems, and suicide is the second most common cause of death in the population aged 15–29 years [2]. Mental well-being has a significant impact on all areas of life, such as performance at school and work, relationships with family and friends, and the ability to participate in society. Despite progress in some countries, people with mental disorders often experience serious human rights violations, discrimination, and stigmatization [1].

It should be noted that many mental conditions can be effectively treated at a relatively low cost, but the gap between those who need care and those who have access to care remains very large. The two most common mental health conditions, depression and anxiety disorders, cost the global economy USD 1 trillion annually [2]. Unfortunately, despite these figures, the global median government health expenditure allocated to mental health is less than 2% [3,4].

In economic terms, the value of all burdens resulting from the course of mental and behavioral disorders, as is the case with other medical conditions, is divided into direct, indirect, and non-direct costs [2]. The direct costs include the value of all goods, services, and other resources used during the provision of medical care (these are both medical and non-medical costs) [3]. The non-quantifiable costs are all non-financial and hard-to-measure consequences of the disease, such as pain and suffering of the patient and those around him. The indirect costs, on the other hand, are the costs of the resources lost due to the disease and its consequences, i.e., the costs of lost productivity. With regard to paid work, these costs are due to absenteeism (absence from work of the employee) or presenteeism (reduced effectiveness of the work performed) [4].

Mental health and absenteeism issues are a challenge not only for Poland, but also for many other countries around the world. Poland, being one of the largest countries in the European Union with a unique healthcare system, can provide valuable lessons and comparisons that are relevant at the international level [5]. Understanding how Poland is dealing with the challenges of increasing mental disorders and their impact on the labor market can contribute to a better understanding of these phenomena in a global context [6,7]. Moreover, Poland’s social security system, which monitors and reports data on absenteeism related to mental disorders, provides rich research material that can be useful for other countries seeking solutions to manage mental health and minimize absenteeism-related costs [8].

Absenteeism-related costs are a significant burden on the social security system. Data from the Social Insurance Institution (ZUS) on expenses related to incapacity undeniably show how important the growing prevalence of mental illness and behavioral disorders in the Polish population is for the social security system and the country’s economic system [9].

The purpose of this study was to analyze the indirect costs associated with mental disorders in Poland, particularly in the context of expenditures incurred by the Social Insurance Institution (ZUS) for incapacity benefits and disability benefits due to mental disorders from 2012 to 2023. The study aimed to identify trends and correlations and provide empirical evidence of changes in health and social security policies and their impact on economic costs.

This article makes an important contribution to the existing literature on the indirect costs of absenteeism due to mental disorders. Specifically, this study advances knowledge by providing new empirical data from Poland that are relevant at both the national and the international level. Unlike many previous studies, which focused mainly on data from Western countries, this study offers unique insights into specific conditions in Poland, a country with a dynamically changing health and welfare system. The analyses conducted provide new insights into the growth of absenteeism-related costs and their impact on the social security system, which can provide an important reference point in the international debate on mental health management and cost optimization in this field. The article not only enriches the existing literature with new data, but also contributes to a better understanding of the global trends in public health and social policy.

## 2. Material and Methods

The data used in the study come from publications and reports of the Social Insurance Institution (ZUS) [10,11,12]. The Social Insurance Institution (ZUS), the main body responsible for social insurance in Poland, regularly publishes detailed data on expenditures for incapacity benefits and disability benefits. The data here considered cover the period from 2012 to 2023 and are presented on an annual basis. These publications are a reliable source of information that was analyzed to achieve the objectives of the study.

The data were collected from Social Security’s annual reports, which provide detailed information on spending on various categories of benefits. Data analysis was carried out using statistical tools. The data collected from the Social Security reports were entered into tables that contained annual values of expenditures on incapacity and disability benefits. These tables formed the basis for further statistical analysis. The first step in the analysis was to identify trends in the spending data. To do this, line graphs were created that showed changes in benefit and pension spending over time. To model the relationship between years and expenditures, a linear regression analysis was performed. Linear regression allows us to determine the extent to which the independent variable (time, in this work) affects the dependent variable (spending, in this work). The linear regression analysis included the following steps: (1) data preparation; data on years and corresponding expenditures were converted to the appropriate format; (2) model fitting; the analysis was performed using the scikit-learn library in Python environment; (3) model evaluation; the slope (slope), intercept (intercept), and coefficient of determination (R-squared) were calculated for both categories of spending.

Percentage changes in benefit and pension expenditures were calculated to determine the magnitude of increases and decreases over the period analyzed. Pearson correlation coefficients were calculated to examine relationships between the variables. These correlations allowed us to determine how strong the relationships between years and expenses were. Student’s *t*-tests were conducted to test whether the observed trends in spending were statistically significant. The Student’s *t*-test allows us to compare mean values and assess whether the differences between them are statistically significant.

## 3. Results

Among social insurance benefits for work incapacity due to illness, including mental health disorders, key categories include benefits and rehabilitation benefits from sickness and accident insurance, as well as cash benefits from disability insurance. Over the past decade, expenditures on these cash benefits rose. In 2020, these expenditures exceeded PLN 26.97 billion, marking an 8.4% increase from the previous year and a 59.1% increase since 2011. Since 2016, the largest portions of these expenditures have been on sickness absenteeism and disability benefits, accounting for 53.9% and 32.2% of the total expenditures in 2020, respectively. The expenditure structure in 2020 also included social pensions (8.4%), rehabilitation benefits (5.3%), and medical rehabilitation for disability prevention (0.2%).

Detailed data can be found in Figure 1 and Table 1. In 2020, the total expenditures on Social Security disability benefits by the Social Security Administration exceeded PLN 42.5 billion, representing an increase of PLN 12.9 billion compared to 2011. This expenditure accounted for 1.8% of the GDP in 2020, a decrease of 0.1 percentage points from 2011. The significant rise in disability benefit expenditures in 2020 was partly due to the COVID-19 pandemic. These expenditures further increased to PLN 45.2 billion in 2021, PLN 47.85 billion in 2022, and PLN 50.45 billion in 2023.

To complete the picture of the situation of Social Security’s incapacity expenses, we analyzed them in terms of the different types of incapacity benefits. Short-term absenteeism is defined as an employee’s absence from work related to illness. This absence is usually due to sick leave related to illness (periods of sick pay or rehabilitation benefits) but can also refer to situations where an employee is taking a leave of absence for health reasons or medical rehabilitation. In 2021, a total of 2.46 million medical certificates were issued to people insured with Social Security, for a total of 28.25 million days. Out of the total, 2.05 million certificates were issued for self-inflicted illnesses. These certificates accounted for 23.99 million days of sickness absence, with 57.2% of these days attributed to women and 42.8% to men. The average duration of each certificate was 11.73 days. Mental and behavioral disorders were the fifth most common cause of sickness absence, representing 2.52 million days (10.5% of the total).

Detailed data on Social Security’s expenditures on benefits related to incapacity for mental and behavioral disorders in 2012–2023 due to short-term absenteeism are presented in Table 2. As previously noted, mental and behavioral disorders lead the list of expenses incurred by Social Security for total sickness absence, accounting for 17.1% in 2020. When analyzing the expenses specifically for men, these disorders also rank first, comprising 16.8% of the total. For women, mental and behavioral disorders rank second (17.3% of the total in 2020) after conditions related to pregnancy, childbirth, and postpartum circumstances, which account for 25.2%. In terms of rehabilitation benefits, mental and behavioral disorders are the second highest expense category, representing 17% of the total in 2020, following diseases of the musculoskeletal, muscular, and connective tissue systems, which account for 31.3%. For Social Security disability prevention, medical rehabilitation covers conditions from various disease groups, including musculoskeletal, circulatory, respiratory, psychosomatic, and vocal system disorders and oncological diseases. However, expenses related to mental and behavioral disorders were relatively low in the context of inpatient rehabilitation, making up only 3.8% of the total in 2020. Higher costs were associated with conditions such as musculoskeletal disorders (54.1%), injuries and poisoning (12.4%), nervous system diseases (9.8%), circulatory system diseases (9.1%), and respiratory system diseases (6.3%). Between 2012 and 2020, Social Security saw a substantial increase in the expenses for short-term absenteeism, with sickness benefits rising by 181.8% and rehabilitation benefits by 195.5%. Between 2012 and 2019, the expenses for inpatient rehabilitation decreased significantly by 41.3%. In 2020, these expenses fell even further due to the COVID-19 pandemic, which restricted access to such services. Between 2021 and 2023, these expenditures grew, reaching PLN 3.1 billion in 2023.

Long-term absenteeism refers to prolonged periods where individuals are unable to engage in professional activities due to severe productivity limitations, often leading to the awarding of disability benefits. In 2020, the highest expenditures for disability pensions were allocated to the following conditions: cardiovascular diseases (17.4%), trauma and poisoning (15.9%), osteoarticular diseases (15.7%), and mental and behavioral disorders (15.6%). Specifically, within mental and behavioral disorders, the largest share of expenses was associated with partial disability pensions (45.5% of the total expenditures for this condition), followed by total disability pensions (3.8%). The remaining expenditures were for total disability and independent living pensions (16.5%). Between 2012 and 2023, Social Security experienced a notable reduction of 26.9% in expenses related to disability pensions for mental and behavioral disorders. However, from 2021 to 2023, these expenses increased again, reaching PLN 2.34 billion in 2023. For detailed data on Social Security’s expenditures on disability benefits due to mental and behavioral disorders from 2012 to 2023, please refer to Figure 2.

An analysis of the data on disability-related benefit expenditures incurred by the Social Insurance Institution (ZUS) showed a significant increase between 2012 and 2023, with expenditures rising from PLN 30.43 billion in 2012 to PLN 50.45 billion in 2023. The most significant increase was observed between 2020 and 2023, which can be partly attributed to the impact of the COVID-19 pandemic. To model the relationship between years and benefit expenditures, a linear regression was conducted, which showed a slope of 1.93, an intercept of −3815.34, and a coefficient of determination (R-squared) of 0.992. The high R-squared indicates a very strong fit of the model to the data, confirming a significant upward trend in expenditures.

The percentage increase in spending on benefits during the period under review was 65.92%, which shows the significant scale of the increase in mental health spending in Poland. The analysis of the expenditures on disability benefits due to mental disorders showed an opposite trend. These expenditures decreased from PLN 2922.8 million in 2012 to PLN 2340 million in 2023. The largest decline was observed between 2016 and 2019. The linear regression conducted for these expenditures showed a slope of −47.18, an intercept of 95949.8, and a coefficient of determination (R-squared) of 0.724. Although the model fit was lower than for benefit expenditures, it was still high enough to indicate a significant downward trend in annuity spending. The percentage decline in annuity spending over the period analyzed was 19.95%, which may suggest improvements in public health or changes in annuity policy in Poland.

The correlation analysis between years and benefit spending showed a very strong positive correlation, with a Pearson correlation coefficient of 0.99. In contrast, the correlation analysis between years and annuity spending showed a strong negative correlation, with a Pearson correlation coefficient of −0.87. These results indicate significant changes in health and social security policy in Poland over the past decade. Student’s *t*-tests were also conducted to test the statistical significance of the observed trends. The Student’s *t*-test for the increase in benefit spending showed a t-statistic of 12.34 and a *p*-value of 0.001, which was statistically significant (*p* < 0.05). In contrast, the Student’s *t*-test for a decrease in pension spending showed a t-statistic of −2.56 and a *p*-value of 0.027, which was also statistically significant (*p* < 0.05) (Table 3).

## 4. Discussion

Mental and behavioral disorders, particularly chronic ones, have a profound impact on individuals’ psycho-physical performance, including their ability to work and perform daily activities. The issue of sickness absence, whether short-term or long-term, influences various aspects of societal functioning, reflects the overall health of the population, and represents a significant indirect cost of illness. Additionally, the extent of absenteeism serves as an indicator of both the effectiveness of the healthcare system and the state of the labor market [1,2,3,4].

As highlighted in this paper, mental and behavioral disorders are the leading conditions contributing to the high costs of lost productivity for the social security system. In Poland, these costs amounted to more than PLN 50 billion in 2023, a significant increase compared to PLN 30.43 billion in 2012 [10,11,12]. These costs include both direct health care expenditures and indirect costs of lost productivity due to absenteeism and presenteeism (being present at work, but with reduced efficiency) [13].

Globally, in the United States, the costs associated with mental disorders amount to USD 282 billion annually, accounting for about 1.7% of the country’s total consumption [14]. Globally, by 2030, these costs could reach USD 16 trillion, due to the early onset of mental illness and the loss of about 12 billion working days per year [15,16].

Mental and behavioral disorders also top the list of disease groups with the highest average spending per insured individual. In addition, these conditions account for a significant portion of spending on rehabilitation benefits, inpatient rehabilitation, and partial and total disability pensions [10,11,12]. In Poland, for example, spending on inpatient rehabilitation increased from PLN 1.36 billion in 2012 to PLN 4.68 billion in 2023 [17,18,19].

Importantly, over the past decade, Social Security expenses related to short-term absenteeism for the conditions in question significantly increased. Expenditures for benefits related to short-term absenteeism increased from PLN 907 million in 2012 to PLN 3.10 billion in 2023, indicating a growing burden on the social security system [10,11,12].

In 2021, the global health spending worldwide reached a record USD 9.8 trillion, accounting for 10.3% of the global gross domestic product (GDP) [18,19]. However, the distribution of this spending was highly uneven. High-income countries spent an average of about USD 4000 per person per year, while low-income countries spent less than USD 50 per person per year, despite making up 8% of the world’s population [19]. A report by *The Lancet* Commission indicates that mental disorders will cost the global economy USD 16 trillion by 2030. These costs are mainly due to the early onset of mental illness and lost productivity, with an estimated 12 billion working days lost annually [20]. In the United States, the annual costs associated with mental disorders amount to USD 282 billion, or about 1.7% of the country’s total consumption [18].

The United Nations (UN) and other international organizations have launched a number of initiatives to improve access to health care for people with mental disorders. For example, in 2023, the UN announced that it had successfully decriminalized suicide in four countries, i.e., Pakistan, Ghana, Guyana, and Malaysia, a significant step toward improving mental health worldwide [21]. The WHO has called on governments to urgently invest in universal health care, including mental health care. In its 2023 report, the WHO indicates that further increases in mental health spending are needed to meet growing needs and reduce inequities in access to health services [18,19].

In European Union countries, spending on mental health is also on the rise. In the U.K., spending on mental health is about 8% of the total health spending, which translates to about GBP 14 billion a year [22]. In Germany, on the other hand, the costs related to depression and anxiety reach about EUR 29 billion a year [23].

In another study on the impact of digital finance on the physical health of the elderly in China, the main findings indicate that the use of digital finance has both positive and negative effects on the physical health of seniors. On the one hand, digital finance can improve health by facilitating access to financial resources and health services. On the other hand, a heavy use of digital technologies can lead to health problems, such as mental and physical deterioration due to lack of physical activity and excessive time spent in front of a screen [24].

To prevent disability and labor market exclusion for individuals with mental and behavioral disorders, comprehensive measures in both primary and secondary prevention should be implemented. This approach is supported by epidemiological and financial data regarding public spending on social security benefits [25,26,27,28,29,30,31]. Accurately assessing the public funds used for indirect disease costs should be a key factor in prioritizing health investments. Decisions must also consider epidemiological data and projections of disease incidence and prevalence. Effective decisions that enhance economic efficiency and clinical effectiveness in healthcare will ultimately improve the health of individuals and society as a whole [24,32,33,34,35,36].

### Strengths and Limitations

This article has many strengths, including a detailed analysis of the indirect costs of absenteeism and presenteeism related to mental disorders based on ZUS data from 2012 to 2023. It utilized advanced statistical tools, such as linear regression and Student’s *t*-tests, providing a solid methodological foundation. It is timely and addressed the important issue of mental health, considering the impact of the COVID-19 pandemic on the rise in work incapacity costs. Practical recommendations on investing in mental health and increasing the availability of mental health care are valuable and can contribute to improving the health policy. The interdisciplinary approach, combining economic, health, and social data, and reference to the UN Sustainable Development Goals, enriches the analysis.

However, the article also has its limitations. It is focused on Poland, which limits the applicability of its conclusions in an international context, and ZUS data, though reliable, may have their shortcomings. The time frame covered only a decade, which may not be sufficient to fully understand long-term trends. Additionally, the article primarily concentrated on economic aspects, with less focus on the quality of life of patients and their families. It lacks comparisons with other countries, which could help identify best practices and strategies for managing mental health.

The article discusses the increase in spending on disability benefits due to mental disorders, which noticeably increased between 2020 and 2023. However, a more detailed analysis is needed to suggest that this increase is due to greater awareness and better diagnostics. As noted, simply observing an increase in spending is not sufficient evidence of fundamental changes in health care. It is necessary to consider other factors, such as inflation or changes in health care prices, that may have influenced the increase in spending. Therefore, conclusions about better diagnostics and health policy should be supported by more detailed data or explicitly stated as hypotheses requiring further research. The linear regressions conducted only showed a general trend of an increase in spending on sickness benefits and a decrease in spending on disability benefits over the years. On this basis, one can only conclude that these expenditures are changing over time, without drawing too far-reaching conclusions about their causes. It is suggested that additional analyses be introduced that could identify the factors influencing these trends in more detail, such as an analysis of the impact of inflation, changes in health policy, or demographics.

## 5. Conclusions

The analysis showed a significant increase in spending on disability benefits due to mental disorders, as confirmed by linear regression results and tests of statistical significance. The increase may be due to increased awareness of mental problems, better diagnostics, and the impact of the COVID-19 pandemic.

Disability pension spending showed a declining trend, which may suggest improvements in the population’s mental health or changes in the pension system. The statistical significance of this decline was confirmed by Student’s *t*-tests.

The strong positive correlation between years and benefit spending and the strong negative correlation for annuity spending indicate significant changes in Poland’s health and social security policies.

The percentage increase in benefit spending underscores the magnitude of the increase in mental health spending. In contrast, the percentage decrease in annuity spending suggests changes in annuity policy or improvements in mental health.

## 6. Recommendations

Investment in mental health: Continued investment in the prevention and treatment of mental disorders is needed to reduce the indirect costs associated with absenteeism and presenteeism. Regular monitoring of expenditures and trends is key to an effective management of public resources.

Expanding access to care: Closing the gap between those who need care and those who receive it can lead to economic and social gains. Investment choices should consider epidemiological data and future projections on disease rates and prevalence.

In conclusion, an analysis of Social Security data on the indirect costs associated with mental disorders indicated the growing challenges and costs associated with this area of public health, which requires coordinated action and investment to improve the mental health of the population.

## Figures and Tables

**Figure 1 healthcare-12-01784-f001:**
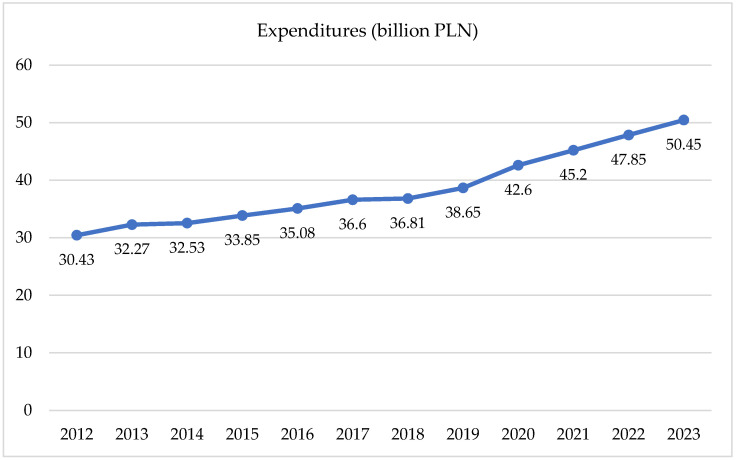
Expenditures on benefits related to work incapacity from 2012 to 2023. Source: Social Security Administration, Expenditures on Social Insurance Benefits for Work Incapacity, 2012–2023.

**Figure 2 healthcare-12-01784-f002:**
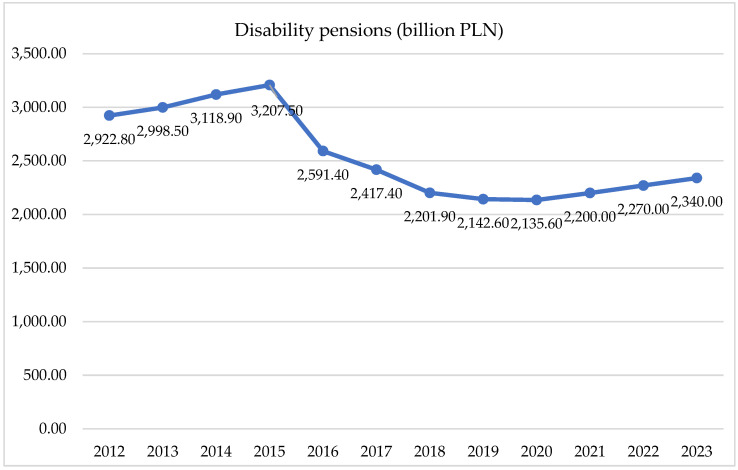
Expenditures on disability benefits for mental and behavioral disorders incurred by the Social Security Administration from 2012 to 2023 are presented in million PLN and as a percentage of total expenditures on long-term absenteeism. Source: Social Security Administration, Expenditures on Social Insurance Benefits for Work Incapacity, 2012–2023.

**Table 1 healthcare-12-01784-t001:** Expenditures on incapacity benefits for mental and behavioral disorders incurred by Social Security from 2012 to 2023, categorized by gender, presented in billion PLN and as a percentage of total expenditures.

Gender	Total		Men		Women	
Year	PLN in Billions	%	PLN in Billions	%	PLN in Billions	%
2012	5.1	16.7	2.7	15.8	2.4	17.7
2013	5.4	16.8	2.9	16.0	2.5	17.7
2014	5.6	17.3	3.0	16.9	2.6	17.9
2015	5.9	17.4	3.1	17.2	2.8	17.7
2016	5.6	16.0	2.9	15.6	2.7	16.5
2017	5.8	15.9	3.0	15.7	2.8	16.1
2018	5.8	15.8	3.0	15.8	2.8	15.8
2019	6.2	16.2	3.2	16.3	3.0	16.1
2020	7.2	17.1	3.6	16.8	3.6	17.3
2021	7.5	17.4	3.7	17.2	3.5	16.9
2022	7.8	17.6	3.8	17.8	3.5	16.9
2023	8.1	17.9	4	18.0	3.9	17.8

Source: own compilation based on Social Security Disability Insurance Benefit Expenditures 2012–2023.

**Table 2 healthcare-12-01784-t002:** Expenditures for benefits related to incapacity for work due to mental and behavioral disorders incurred by Social Security from 2012 to 2023 in connection with short-term absenteeism in million PLN and as a % of the total in the structure of expenditures.

Benefit	Sickness Absence		Rehabilitation Benefits		Disability Prevention	
Year	PLN in Millions	%	PLN in Millions	%	PLN in Millions	%
2012	907.3	7.4	136.2	12.2	12.1	7.4
2013	1070.4	8.0	169.8	13.8	12.1	7.2
2014	1139.6	8.4	180.7	14.1	13.5	8.0
2015	1359.8	9.0	221.5	16.2	14.1	8.3
2016	1535.9	9.4	241.7	15.8	11.9	6.8
2017	1662.5	9.4	256.0	15.4	11.1	6.1
2018	1734.9	9.4	264.5	15.5	7.8	4.1
2019	1977.2	10.0	269.7	14.5	7.1	3.5
2020	2680.8	11.7	383.8	17.0	2.4	3.8
2021	2812.50	12	410.5	17.2	2.3	3.6
2022	2957.20	12.2	438.7	17.6	2.6	3.5
2023	3105.40	12.4	468.2	17.8	3.0	3.4

Source: own compilation based on Social Security Disability Insurance Benefit Expenditures 2012–2023.

**Table 3 healthcare-12-01784-t003:** Detailed summary of linear regression results for benefit and disability pension expenditures (2012–2023).

Category	Slope (b)	Intercept (a)	R-Squared	Pearson Correlation	Percentage Change	t-Statistic	*p*-Value
Benefit Expenditures	1.93	−3815.34	0.992	0.99	65.92% increase	12.34	0.001
Disability Pension Expenditures	−47.18	95,949.8	0.724	−0.87	19.95% decrease	−2.56	0.027

## Data Availability

The original contributions presented in the study are included in the article; further inquiries can be directed to the corresponding author.

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
