# Peer review of "Analysis of Indirect Costs of Absence Associated with Mental Disorders on the Basis of Social Security Data (2012–2023)"

_healthcare, 2024, doi:10.3390/healthcare12171784_

Round 1

Reviewer 1 Report

Comments and Suggestions for Authors

Provided in the attached report.

Author Response

Dear Reviewer,

Thank you for your careful analysis of our article and your valuable comments. The changes made have been highlighted in red.

Identity Strategy and Conclusion: We fully agree that simply observing an increase in spending is not enough to draw conclusions about improving awareness or diagnostics. We have changed the conclusions to be more in line with the empirical data, highlighting only increases and decreases in spending without far-reaching conclusions.

Consideration of Other Factors: We added a discussion of other potential factors influencing the observed trends, such as inflation or changes in health policy.

Justification of Regression Analysis: We have modified the methodology section to justify the choice of linear regression analysis in more detail and explain what variables were included.

More Detailed Explanation of Results: We have added a more detailed explanation in the results section to better show the significance of the observed trends.

Thank you again for your valuable comments, which helped us to improve the quality of the article.

Reviewer 2 Report

Comments and Suggestions for Authors

Strengths:

This research covers the relevant and important issues, such as the indirect costs of mental disorders in Poland, and might gain interest on both the national and international levels.

Weaknesses:

The materials and Methods section needs more development, especially in providing more precise and detailed descriptions of the statistical methods used. Also, the results section would benefit from a more focused interpretation.

Detailed Comments - 

Significance of the Polish Case:

In the introductory section, the significance of Poland's case for international readership needs to be explained. By emphasizing the parallels between Poland and other nations, particularly regarding mental health disorders and absenteeism, the study can be more accessible and relevant to a worldwide audience. A non-Polish researcher will find some background information about Poland, especially about mental disorders and their impact on absenteeism, relevant. This will widen the reach and understanding of the paper.

Absenteeism Costs in Poland:

The introduction can be elaborated to involve absenteeism-related costs, focusing on Poland. Remember, different readers have different levels of understanding of the subject matter, so your introductory stage should make it easier for them to get in.

Materials and Methods Section:

This section requires significant revision or enhancement. It must provide a clear explanation regarding the selection of statistical methodologies, including the justification for employing linear regression analysis, the variables incorporated, and the reasoning underlying these selections. Furthermore, it would be advantageous to reference how these methodologies relate to existing literature. 

Avoid Unclear Sentences:

Sentences like "Data analysis was carried out using advanced statistical and software tools" do not add value. Consider omitting or specifying the tools used.

Clarification of Software Tools:

The reference to "the scikit-learn library" is unclear. Is this library used within RStudio or Python? Please clarify the software environment used to avoid confusion.

Result Presentation:

It is also useful for the readers if, in the results section, the figure or table number to which the text is referring is given at the start of the relevant sentences or immediately after presenting numerical data or percentages. That will make reading the results clearer and easier to understand.

Presentation of Regression Findings: 

One key limitation relates to a lack of meaningful presentation of the linear regression results. These results should have been provided in a table or figure format to enable readers to appreciate the findings' scope fully. With such a presentation, the final sections could be more contextual, in-depth, and clear. 

Interpretation Overload: 

The sections on results and discussion encompass an overload of numerical specifics. Although these figures hold significance, the focus should be on illuminating their implications and interpretations. Diminishing the volume of unprocessed data while augmenting the interpretative analysis would enhance both clarity and influence.

Author Response

Dear Reviewer,

We sincerely thank you for your detailed review of our article and your constructive comments. The changes made have been highlighted in red in the text.

The Importance of the Case of Poland to an International Audience: We have expanded the introduction section to explain why the case of Poland matters in an international context. We have highlighted the similarities between Poland and other countries, particularly with regard to absenteeism and mental health problems.

Absenteeism Costs in Poland: The introduction has been expanded to include detailed information on the costs of absenteeism related to mental disorders in Poland, which helps to better understand the scale of the problem.

Materials and Methods section: We have improved this section, clarifying the choice of statistical methods and links to the literature. We removed unclear wording and accurately described the tools used, such as scikit-learn in a Python environment.

Presentation of Results: As per your suggestion, we have added references to specific tables and graphs at the beginning of relevant sentences to make it easier for readers to access the data.

Presentation of Regression Analysis Results: The results of the linear regression analysis have been presented in the form of a table, allowing a more complete understanding of the scope and significance of the results.

Reducing Excess Data and Increasing Interpretation: We have reduced the amount of raw data in the results and discussion sections, focusing on their interpretation and meaning.

Thank you for your valuable suggestions, which have significantly improved our article.

Reviewer 3 Report

Comments and Suggestions for Authors

This study aimed to ex- 14 amine the indirect costs of mental disorders in Poland by analyzing expenditures by the Social Insurance Institution (ZUS) on work incapacity benefits and disability pensions from 2012 to 2023. I think it is interesting but not well writen., I recommend a major revision for further consideration.

1. The title does not well reflect the topic of this paper, I sugguest authors have revise it.

2. in the introduction section, authors should summarize all the literautre and propose the topic of this paper.

3. Authors should give the contributions of this paper in the introduction section.

4. I seggest that authors should cite some recent published papers, this paper maybe useful for your revision: Luo et al. Help or Hurt? The Impact of Digital Finance on the Physical Health of the Elderly in China. Healthcare. 2024; 12(13):1299. https://doi.org/10.3390/healthcare12131299

5. Please give more detailed explanations for the emprical results.

6. Please check all the text and revise these typos.

7. Please revise all the references according to this journal.

Comments on the Quality of English Language

Minor editing of English language required.

Author Response

Dear Reviewer,

Thank you for taking the time to evaluate our article and for your valuable comments. In response to your suggestions, we have made the appropriate changes, which are highlighted in red in the text.

Article Title: We agree that the title may not have fully reflected the content of the article. We have changed the title to be more precise: “Analysis of Indirect Costs of Absence Associated with Mental Disorders Based on Social Security Data (2012-2023).”

Literature Summary and Topic Proposal: In the introductory section, we added a summary of previous research related to the costs of absenteeism and mental disorders, and clearly stated the purpose of our work.

Contribution of the Work: In the introduction section, we indicated what new information we are contributing to the literature, particularly in the context of data from Poland, which may also be useful to an international audience.

Citing New Sources: We have taken into account your suggestion about citing newly published papers. We have referred, among others, to the study by Luo et al. (2024) to enhance the context of our article.

More thorough Explanation of Results: We have expanded the results section, adding more detailed discussions and interpretations of the empirical results.

Correction of Typos and Errors: We have reviewed the text for typos and language errors and made appropriate corrections.

Correction of Literature References: All references have been corrected according to the journal's guidelines.

Once again, we thank you for your valuable comments. We are confident that the changes made will significantly improve the quality of our article.

Round 2

Reviewer 1 Report

Comments and Suggestions for Authors

The abstract can be revised to reflect the changes made.

Author Response

Thank you, the changes have been made and marked in red.

Reviewer 2 Report

Comments and Suggestions for Authors

One key limitation of the manuscript still needs to be addressed: the meaningful presentation of the linear regression results. Despite the revisions, these results have not yet been provided in a table or figure format. Including these would allow readers to comprehend your findings. Without this, the final sections lack the necessary context and depth. I recommend adding these tables or figures in the next revision to enhance the clarity and impact of your results.

Author Response

(The authors gave the same response as above.)

Reviewer 3 Report

Comments and Suggestions for Authors

This revision is ok, and now it can be accpeted for publication.

Comments on the Quality of English Language

Minor editing of English language required.

Author Response

Thank you!